# Beyond Trees: Regulons and Regulatory Motif Characterization

**DOI:** 10.3390/genes11090995

**Published:** 2020-08-25

**Authors:** Xuhua Xia

**Affiliations:** 1Department of Biology, University of Ottawa, Ottawa, ON K1N 6N5, Canada; xxia@uottawa.ca; 2Ottawa Institute of Systems Biology, Ottawa, ON K1H 8M5, Canada

**Keywords:** gene expression, transcription factor, regulon, regulatory motifs, comparative genomics, Gibbs sampler

## Abstract

Trees and their seeds regulate their germination, growth, and reproduction in response to environmental stimuli. These stimuli, through signal transduction, trigger transcription factors that alter the expression of various genes leading to the unfolding of the genetic program. A regulon is conceptually defined as a set of target genes regulated by a transcription factor by physically binding to regulatory motifs to accomplish a specific biological function, such as the CO-FT regulon for flowering timing and fall growth cessation in trees. Only with a clear characterization of regulatory motifs, can candidate target genes be experimentally validated, but motif characterization represents the weakest feature of regulon research, especially in tree genetics. I review here relevant experimental and bioinformatics approaches in characterizing transcription factors and their binding sites, outline problems in tree regulon research, and demonstrate how transcription factor databases can be effectively used to aid the characterization of tree regulons.

## 1. Introduction

Research in tree genetics, especially in gene regulation networks, has benefitted much from looking beyond trees. The finding of animal c-MYB family of genes resulted in the discovery of the R2R3-MYB family of transcription factors in *Arabidopsis thaliana* [1,2], and the understanding of how *A. thaliana* responds to cold through gene regulation sheds light on how tree species such *Malus demestica* respond to cold [3]. I present a conceptual framework here to highlight some shortcomings in tree regulon studies to facilitate better experimental designs in the future.

The term regulon was originally proposed [4] as an extension to operon, in the context of a repressor controlling the expression of multiple genes that are not located within the same operon. For example, the eight genes involved in the biosynthesis of arginine are scattered in five separate genomic locations in the genome of *Escherichia coli*, but controlled by the same arginine repressor [4]. Since its early usage, “regulon” has been associated with specific biological processes, e.g., the *argR* (the *E. coli* gene encoding the arginine repressor) regulon for arginine biosynthesis, or the CO-FT regulon for flowering timing and seasonal growth in *Arabidopsis thaliana* [5] and in trees [6].

The adoption of *Arabidopsis thaliana* as a model for research in plant molecular biology has resulted in the discovery and characterization of regulons in *Arabidopsis thaliana* and related plants, but also in various tree species. Some regulons function similarly, such as the CBF regulon in plant response to low temperature [3,7,8] which is gated by a circadian clock in both *Arabidopsis thaliana* [9,10] and peach trees [11]. However, some other regulons appear to function differently. For example, the CO-FT regulon controls genes involved in seasonal growth and flowering time in *Arabidopsis thaliana* [5], but flowering and growth cessation appear to be two different processes in *Populus* species. While flowering time is still under the control of the CO-FT regulon [6], seasonal growth cessation is regulated by an additional set of genes [12].

Many different experimental and bioinformatics methods have been used to discover and characterize regulons, but the large-scale approach of combining ChIP-seq and RNA-seq technology has been the most frequently used in recent years [13,14] evolving from the microarray method that had been used successfully in the characterization of the CESA regulon for cellulose biosynthesis [15,16]. New derivatives from ChIP-seq, such as ChIP-exo [17,18] ChEC-seq [19] and Cut&Run [20], as well as new technologies such as DamID-seq [21], have been developed in recent years. I evaluate these experimental methods and bioinformatics algorithms after a formal definition of regulon and regulon network.

## 2. A Formal Definition of Transcription Regulon and Regulon Network

A conceptually ideal transcription regulon should have four features: (1) a transcription factor, (2) one or more target genes whose expression is regulated by the transcription factor, (3) a set of conserved transcription factor binding sites (TFBS) on the target genes, and (4) specific biological functions accomplished by altering the expression of the target genes. All regulons I mention in this review are transcription regulons.

The importance of feature 3 above may be illustrated in Figure 1. Suppose a transcription factor A (tfA) can bind to TFBS X and regulate three target genes, *T1*, *T2* and a transcription factor B (*tfB*), that share TFBS X. tfB protein can in turn bind to TFBS Y to control the expression of target genes *T4* and *T5* that share TFBS Y in their regulatory region. Criterion 3 implies that the tfA-regulon includes target genes *T1*, *T2,* and *tfB*, but not *T4* and *T5* whose TSBS Y is bound by tfB but not by tfA. In contrast, the term “coregulated genes” would typically include *T1, T2, tfB, T4,* and *T5* as coregulated directly by tfA or indirectly by tfA through tfB.

In rare cases, two different TFs represent nearly identical paralogues and feature the same DNA- binding affinity. Examples include *MSN2* on chromosome 13 and *MSN4* on chromosome 11 of the yeast *Saccharomyces cerevisiae* [22], and *MdMYB124* (*MYB88*, transcription factor MYB51) on chromosome 13 and *MdMYB88* (*LOC103402919*, transcription factor MYB88) on chromosome 16 of the apple [3] shown in Figure 1D. They should be designated as MSN2/MSN4 regulon and MdMYB124/MdMYB88 regulon, respectively. Such redundancy may enhance regulon reliability in responding to critical environmental stimuli [22].

Regular research aims to characterize the four features of a regulon and to understand its biological function. Among the four features, TFBS is typically the worst characterized, partly through negligence and partly through poor experimental design. For example, the partial regulon network Figure 1D) for developing cold hardiness in response to cold stress does not feature established TFBS because, although the researchers [3] took the ChIP-seq approach, the ineffective experimental design thwarted the effort. I discuss this in more detail later.

Feature 1 of the regulon definition is essential for us to understand the dynamic nature of regulon operation. A transcription factor does not bind to TFBS of its target genes all the time, but only at specific time under specific environment, e.g., CO-FT regulon in response to seasonal change in photoperiod [5] or bZIP60 regulon in response to unfolded protein response (UPR) in endoplasmic reticulum [23,24]. This problem is particularly serious for lowly or transiently expressed TFs, and it is partly for this reason that the DAP-Seq approach with TFs highly expressed in vitro can often identify more TFBSs than alternative methods [25]. Thus, the arrow of protein-DNA binding results from collapsing over different time and intracellular/extracellular conditions, and consequently is not useful in predicting dynamic interaction of the regulon. Knowing the function of the regulon helps us understand when gene regulation should take place.

The definition of transcription regulon above does not include gene regulation mediated by mechanisms other than transcription factor binding to TFBSs, i.e., it does not include gene regulation involving posttranscriptional or posttranslational modification, DNA methylation, or histone modification. For example, yeast *HAC1* mRNA is exported into cytoplasm in an unspliced form, and is translated to produce functional proteins only after the *HAC1* mRNA is spliced by Ire1 to remove the translation inhibition mediated by its intron binding to its 5’ UTR [26,27,28] However, we do not consider Ire1-Hac1 as a regulon because the regulation is not achieved by Ire1 binding to the TFBS of *HAC1* to regulate the transcription of *HAC1.* A regulon could have just a single gene. For example, *HAC1* is a transcriptionally autoregulated gene with its protein binding to its TFBS forming a positive feedback loop until Ire1 stops splicing *HAC1* mRNA [29]. Thus, a transcriptionally autoregulated gene, with its protein regulating the expression of its own mRNA, can be a regulon without involving any other genes.

A regulon network involves two or more interacting regulons. The interacting tfA-regulon and tfB-regulon (Figure 1) constitute a regulon network. In this framework, the *CO-FT* regulon regulating flowering time in *Arabidopsis thaliana* [5] and in *Populus* species [6] would be considered a regulon network instead of a regulon. In fact, CO and FT are equivalent to tfA and tfB, respectively, in Figure 1. Another example of a regulon network is the *ICE1–CBF–COR* regulon network [30,31,32] in which ICE1, in response to cold, binds to TFBS of *CBFs* to upregulate their expression, and CBFs bind to TFBS of their downstream cold-regulated genes (CORs) to enhance cold tolerance. The key difference between *ICE1-CBF-COR* regulon network and the regulon network in Figure 1 is that *CBFs* include multiple genes instead of a single *tfB.*

A regulon network, as represented in Figure 1C with the interacting tfA-regulon and tfB-regulon, is a tree with height of 2 (Tree height is the maximum number of edges from the root to the tip). If T4 protein (Figure 1) were a transcription factor regulating its own target genes, then the regulon network would be a tree of height 3. Because biological systems are typically noisy, a robust regulon network represented as a tree should have low height. In other words, if we have a long chain of tfA→tfB→tfC→tfD…, and if each link has a chance of failure, then the reliability of tfA triggering the last transcription factor in the chain is reduced as the chain length (tree height) is increased. However, information propagation in biology is rarely through a single chain but more often through redundant pathways to increase robustness.

A regulon network (RN) differs from the conventional gene regulation network (GRN) in the basic unit of the network. A basic unit in RN is a regulon with its four features (TF, TFBS, target genes, and biological function), but that in a GRN is not explicitly defined, although two nodes in GRN connected by a directed or undirected line are assumed to be in regulator–regulatee relationship. An RN highlights which feature of a regulon is missing. For example, Figure 1D shows clearly the missing TFBS without which one cannot scan the genome for candidate target genes and verify the regulator–regulatee relationship.

While some regulons in tree species are well characterized [3,6,12], the regulon definition above is best exemplified by the Hac1-regulon in yeast *Saccharomyces cerevisiae* [33,34]. First, Hac1 is a well-characterized master transcription regulator with a basic-leucine zipper motif that controls the initiation of unfolded protein response triggered by an abnormal accumulation of unfolded/misfolded proteins in the endoplasmic reticulum (ER) [27,35,36,37]. Second, the regulator has a fairly well documented set of 381 target genes [38] serving two key functions related to restoring ER homeostasis: (1) increasing the folding capacity by upregulating the ER-resident chaperone proteins [39], and (2) decreasing folding demand by selectively degrading unfolded/misfolded proteins [38,40], extracellular export of unfolded/misfolded proteins [41], and reduction of protein synthesis [42]. Third, it features two well-validated TFBSs (unfolded protein response element, UPRE): UPRE1 (GACAGCGTGTC) [35,39,43,44] and UPRE2 (TACGTGT) [38,39,45,46] that is present in target genes, e.g., *KAR2* [35]. Fourth, its function within UPR is well defined and specific, i.e., to restore ER homeostasis or, if the restoration fails, pass the signal to other cellular machinery to trigger apoptosis.

Multiple approaches have been used to verify the association between TF (e.g., Hac1) and TFBS of target genes (e.g., *KAR2*). The first is to check if TF fails to bind to, and regulate, the target gene after removing or mutating the putative TFBS [35], or TF-binding occurs when the putative TFBS is inserted into another gene that does not originally have a TFBS and does not bind to TF. This cannot be done in large-scale and is done less frequently today. Second, if target genes T1, T2, and T3 harbor this TFBS, then their expression should differ between the wild type with a functional TF and a mutant without the TF. This is typically done as part of a ChIP-Seq experiment. One such experiment is exemplified by BioProject PRJNA633509 in the NCBI SRA (Sequence Read Archive) database, with the objective to characterize transcription factor narrow sheath1 (NS1), its target genes, and its binding sites in *Zea mays*. The ChIP-Seq data set would allow one to identify putative TFBSs in a large number of putative genes. Transcriptomic data were obtained from the wild type with a function NS1 and an *ns1* mutant. A gene is likely NS1-regulated if it not only harbors a putative TFBS, but also differs in gene expression between the wild type and the mutant. The third approach is to compare sequence conservation of orthologous genes among related species. A TFBS should be strongly conserved but its flanking sequences should not. These approaches have been used to characterize the two UPREs above and the target genes of Hac1.

That Hac1 can bind to two separate motifs is particularly relevant to regulon characterization and interpretation of experimental results. Yeast researchers were once puzzled by the observation that many genes whose transcription is driven by Hac1 do not have UPRE1 [38,45,46] Instead, they appear to share a different motif, later named UPRE2. UPRE1 was an experimentally validated TFBS for Hac1 [35,43,44]. In the conventional wisdom of one specific regulatory motif for one transcription factor, it was hypothesized that another transcription factor was involved, playing the role of tfB in Figure 1, with Hac1 equivalent to tfA (Figure 1). The equivalent of tfB was identified (or perhaps misidentified) as Gcn4 [45], i.e., Hac1 regulates *GCN4* transcription and the resulting Gcn4 regulates the transcription of those yeast genes that do not have UPRE1 but have UPRE2. However, *GCN4* does not have UPRE1, and its transcription remains unchanged during UPR [45], so it cannot possibly be equivalent to tfB in Figure 1. It turns out that Hac1 is capable of binding to either UPRE1 or UPRE2 [39]. This example illustrates the importance of characterizing regulatory motifs of transcription factors. The detailed elucidation of yeast UPR demonstrates the success of long-term, persistent and focused research since early 1990s [33,34]. An equivalent success story in plant sciences is the characterization of *AtHB1* during roughly the same period [47,48,49], although much remains unknown in AtHB1-mediated regulation based on the four-feature criterion of a regulon.

The skeletal scheme in Figure 1 serves only as a starting point for conceptualization of regulons and regulon networks. Gene regulation can occur in unexpected ways. For example, yeast Hac1 can upregulate mRNA expression of a set of target genes involved in UPR, but downregulate protein production of a subset of these genes [50]. It accomplishes this by altering the transcription start site to include a short open reading frame in the upstream of some target genes, so that the resulting mRNA cannot be translated efficiently, leading to reduced protein production. A standardized set of conceptual definitions is need to avoid confusion in regulon research.

## 3. Discovery and Characterization of New Plant Regulons

There are two categories of methods for discovery and characterization of new plant regulons. The first is to check the “regulon dictionary” (transcription factor databases) upon finding a new gene in the same way as we come upon a new word. If the dictionary does not contain the new word, then we make use of the second approach of inferring the meaning of the word based on the context and our grammatical knowledge. This is the approach of de novo discovery of transcription factors based on their shared features, but not their presence in the dictionary. If the new word (gene) turns out to be a transcription factor, then it goes to enrich the regulon dictionary.

### 3.1 Regulon Discovery by Checking the Regulon Dictionary

There are now many databases of characterized transcription factors, from the early TRANSFAC [51] to more recent JASPAR [52], CIS-BP [53], Cistrome DB [54], ChIPBase [55], GTRD [56], the last three all being based on ChIP-Seq data. Plant-specific databases of transcription factors and their TFBS have also become available recently [57,58]. Unfortunately, but as expected, an overwhelming majority of transcription factors in plants are from *Arabidopsis thaliana.* The rest are mostly from crop species. Thus, the field of tree regulon is essentially a virgin land. Note that these databases do not feature an exhaustive compilation of known TFBSs. They may include TFBS from some large-scale studies, but ignore TFBS detected and verified through conventional painstaking experiments. For example, they include one TFBS of yeast Hac1p [38,39,45,46] but not the other (GACAGCGTGTC) that was well-characterized earlier by conventional experiments [35,39,43,44]. Such incompleteness could be misleading. If a new researcher identifies GACAGCGTGTC as a TFBS for Hac1p, he may search the databases for Hac1p TFBS. Upon finding no such an entry, he may be misled to think that he has identified a TFBS new to science.

In spite of the incompleteness, the regulon databases are useful for two main purposes. The first is to check if one’s gene is homologous to a transcription factor already characterized in the database. The second, which should have been more commonly employed, is to retrieve the compiled binding sites to scan new sequences for binding sites [59,60,61], which I illustrate in detail below. Almost all these databases output experimentally characterized regulatory motifs of specific transcription factors in the form of a matrix. The one shown in Figure 2A is for AtHB1 (GenBank locus_tag AT3G01470) as retrieved from PlantPAN [57]. The first three and the last two sites are less informative than sites 4 to 12 (Figure 2B) which may be simplified as a palindromic CAATYATTG. From these nine highly informative sites, one can obtain a position weight matrix or PWM (Figure 2C) with entries specified as Equation (1)
(1)$${\mathrm{PWM}}_{\mathrm{ij}}=\log_{2}\left(\frac{p_{\mathrm{ij}}}{p_{i}}\right)$$
where *p_ij_* is the frequency of nucleotide i (i = A, C, G, or T) at site j, sometimes with pseudocounts to avoid taking logarithm of zero, and *p_i_* is background frequencies. The pros and cons of various pseudocount schemes, and of different background frequencies, for PWM have been reviewed and numerically illustrated in detail [62,63].

The resulting PWM (Figure 2C) can now be used to scan new sequences for candidate TFBS. This is done with a sliding window of nine sites (same as the number of sites in PWM) by computing a window-specific PWM score (PWMS). A 9mer with a high PWMS implies a greater likelihood of being a TFBS. PWM allows us to obtain PWMS of a 9mer by summing up relevant PWM entries. For example, if the nine sites in a window happen to be CAATCATTG, then its PWMS is simply the summation of the bolded numbers in Figure 2C. This would result in a PWMS of 14.30496, the highest PWMS we can obtain given the PWM (Figure 2C). Suppose we use this PWM to scan the 2000 nt upstream of a micro-RNA gene *miR164a*. Ideally, the background frequencies, i.e., *p_i_* in Equation (1), should be the nucleotide frequencies of these 2000 nt upstream of *miR164a* (which are 0.3005, 0.1795, 0.1811, 0.3389 for A, C, G, and T, respectively). These frequencies were in fact used to compute the PWM (Figure 2C). The highest PWMS (=10.2048) was found at site 343 (with *miR164a* starting at site 2001), with a 9mer CAATCATTA. This 9mer matches eight out nine sites in the consensus (Figure 2B) and its PWMS is much higher than those upstream or downstream from this site (Figure 2D). The expected PWMS is zero for sequences assembled randomly from the background frequencies. A PWMS of 10.2048 means that the hypothesis of a site-specific pattern is 1180 times as likely as the hypothesis of no site pattern, i.e., CAATCATTA is highly likely a TFBS for AtHB1. A detailed experimental study [64] showed the 9mer to be indeed a functional TFBS for AtHB1 to regulate the expression of *miR164a* expression. Through such a combination of bioinformatics and experimental studies, the last two features of a regulon (i.e., the regulated target genes and the key regulon function) may be elucidated. A number of programs can scan sequences for putative TFBSs based on an input PWM, and the performance of four such programs has recently been evaluated [65].

The characterized binding sites in transcription databases may also help us determine the presence or absence of direct regulator–regulatee relationship. For example, *AtHB1* is upregulated in hypocotyls and roots of seedlings in response to short day length, and its expression was claimed to be regulated by PIF1/PIL5 [68]. The binding site of PIL5 (locus_tag AT2G20180) is the G-box transcriptional regulatory code (CACGTG) which is shared among many different plant transcription factors [69]. However, there is no CACGTG in the 2000 nt upstream of the *AtHB1* gene. The closest 6mer one can get is CTCGTG at site 781 (*AtHB1* start codon at site 2001) and CACGAG at site 1270. Scanning an additional 1000 nt further upstream does not find better matching. Because TFBS is rarely located more than 3000 nt upstream of a coding sequence, PIL5 is unlikely to be a direct regulator for *AtHB1* expression. If the regulation is indirect through another transcription factor, then PIL5 and AtHB1 may belong to the same regulon network but not the same regulon according to the definition laid out in the previous section. This highlights the importance of using well-characterized regulons to build regulon networks.

### 3.2. De Novo Regulon Discovery

#### 3.2.1. De Novo Discovery of TF

When we encounter a new word, cannot find it in dictionaries, and wish to judge if it is a verb, we have to use our grammatical knowledge. Transcription factors, like verbs, also share certain features that allow them to be identified and classified [70]. All transcription factors feature a nuclear localization signal [71], a specific structure, such as a leucine zipper [72], for binding to TFBS, and a positively charged DNA-binding domain to facilitate favorable electrostatic interaction with the negatively charged DNA backbone.

The nuclear localization signal (NLS) is typically rich in basic amino acids Lys and Arg. The first NLS is PKKKRKV [71]. The NLS is 29RKRAKTK35 in the UPR master transcription factor Hac1 in yeast [73], 75RRKLKNRVAAQTARDRKK92 in XBP1 in human [74], and 645RKRGSRGGKKGRK657 yeast Ire1 [75]. The minimum consensus in mammalian NLS is KR/KxR/K [76].

Transcription factors fall into several major structural classes such as leucine zipper, helix-loop-helix, helix-turn-helix, and many others [70]. Leucine zippers feature their characteristic heptad repeats (Figure 3) with repeated 7mers represented as (abcdefg)_n_. Positions a and d are hydrophobic. In leucine zipper transcription factors such as GCN4 in yeast [77], and XBP1 in human [74], the fourth position in each heptad (position d) is occupied by leucine (Figure 3A,B). Heptad repeats are relatively poor in glycine (which would permit too much bending flexibility). They form helices, contain no helix-breaking proline and few clustered charged residues. Hydrophobic residues at positions a and d are on the same side of the helix (Figure 3C and D) and form a hydrophobic interface in a homodimer. Plants also have a Hac1 functional homologue, e.g., bZIP60 in *Arabidopsis thaliana* [23,24] with a leucine zipper structure. It is significant to note that yeast Hac1p [38,39,45,46] mammalian Xbp1 [78,79] and plant bZIP60 [80] all have a TFBS with a core ACGT, although little sequence homology exists among the three. For example, there is only weak homology in a short stretch of amino acid sequences between Hac1p and Xbp1 corresponding to sites 29–53 in the yeast Hac1p [28]. This short stretch is the nuclear localization signal in Hac1p [73].

In addition to the basic NLS, transcription factors typically feature a basic DNA-binding domain favoring electrostatic interaction with the negatively charged DNA backbone. The domain could be in the middle as in AtHB1 (Figure 4) or at the N-terminal as in Hac1 and XBP1 from yeast and human, respectively (Figure 4). The last two are functional homologues serving as master regulator of UPR, but share no detectable sequence homology at either nucleotide or amino acid level.

While none of the features above can identify a protein as a TF, the presence of all these features jointly does increase the likelihood of a protein being a TF. If a protein not only has all these features, but its expression in the cell also consistently results in altered expression of a set of genes, then we can conclude with reasonable confidence that the protein is indeed a transcription factor. Unfortunately, there is no software at present that integrates all available information in de novo discovery of TFs.

#### 3.2.2. De Novo Discovery of TFBS

There are computational methods for predicting DNA-binding sites from 3D protein structure [81,82,83,84,85] or even from plain amino acid sequences [86], especially from sequence similarity of DNA- binding domains [53], although the reliability of these approaches, especially the latter one without structural information, is questionable. A statistical trend often does not imply accuracy in specific predictions. For example, Weirauch et al. [53] likely have overgeneralized the observation that “closely related DBDs almost always have very similar DNA sequence preferences” to the conclusion that “in general, sequence preferences can be accurately inferred by overall DBD AA identity”. I do not disagree with the observation and can list examples more impressive than they did. For example, TFs such as Hac1p in yeast [38,39,45,46], Xbp1 in mammals [78,79] and BZIP60 in plants [80] all have a leucine zipper DNA-binding structure, and their TFBS all feature the core ACGT. However, I disagree that such observations, remarkable as they are, imply much power in making specific predictions. If I states that “Because transcription factor X has a DBD nearly identical to that of transcription factor Y, I predict that transcription factor X has nearly identical DNA sequence preference as that of transcription factor Y”, few experimental biologists would take my prediction seriously and act upon it. In other words, few people would agree with the assertion that “in general, sequence preferences can be accurately inferred by overall DBD AA identity”, as one could list a number of counter examples in which sequence preferences cannot be accurately inferred by overall DBD AA identity. In short, computational predictions of TFBS, and the resulting databases such as CIS-BP, could be valuable, but one should be aware of their limitations in making specific predictions.

Experimental approaches for large-scale de novo discovery of TFBSs include microarray, ChIP- Seq and its derivatives such as ChIP-exo [17,18], ChEC-seq [19] and Cut&Run [20]. All these methods fragment DNA, either by sonication, by Micrococcal nuclease as in ChEC-seq and Cut&Run, or by lambda exonuclease as in ChIP-exo (Figure 5). The microarray approach will hybridize the extracted DNA fragments to a microarray. Because probe sequences on the microarray are known, binding to a probe means that the probe sequence either contains a candidate TFBS or is very close to it along the genome. Such information can be used to produce a position-specific affinity matrix (PSAM) by using software such as MatrixReduce [87]. This PSAM can then be used to scan new sequences, in the same way as PWM, for candidate TFBS.

The ChIP-Seq approach will sequence all sequence fragments (Figure 5F), each of which may contain no or multiple TFBSs. Because TFBS length is typically in the order of 10, the sequence fragments from ChIP-Seq ideally should be just the plain TFBS without flanking sequences. However, ChIP-seq with sonication often generates sequences of several hundred bases. A common approach is to sequence both ends, knowing that the putative TFBS is located between the two ends (Figure 6A). If the two ends overlap, then the putative TFBS is included in the sequence. If the two ends do not overlap (e.g., when the fragment is 500 base and the paired-end read length is 100 bases), then potentially the two paired ends can be mapped onto the genome to obtain the complete sequence in between. Figure 6B shows two paired-end reads from a Cut&Run experiment on *Mus musculus* (BioProject ACCN PRJNA544746) that do not overlap. By mapping them onto mouse chromosome 2 (NC_000068), we learn that they are separated by just a single nucleotide C (Figure 6C). However, if we consider TFBS as signal and the flanking sequences as noise, then long flanking sequences are associated with low signal/noise ratio. TFBS detection is more sensitive with short flanking sequences as in Figure 6C. One of the advantages of ChIP-exo [17,18] ChEC-seq [19] and Cut&Run [20] is the use of MNase or exonuclease which cuts unbound DNA (e.g., the linker region nucleosomes) and digests the free ends until it encounters the bound (protected) region. The fragment length in Figure 6D is likely the minimum achievable through sonication. Most sequence reads by ChIP-seq methods are much longer than that in Figure 6D. For example, in an effort [3] to obtain TFBS in two paralogous TFs (MdMYB88 and MdMYB124) in apple (*Malus domestica*), sonication was used to generate sequence fragments of 300–700 bp, but the single-read has a length of only 50 bases. This implies that, if TFBS is in the middle of each sequence fragment, then the read of 50 bases from the sequence fragment will not include TFBS, so the signal/noise ratio is 0. However, with MNase or other exonucleases, it is theoretically possible to generate TFBS-containing sequence fragments of only 50 bases or even shorter.

With a large number of sequence reads containing the putative TFBS for a particular TF, there are two categories of bioinformatics approaches to extract TFBS. The first, represented by MACS [88], is to map the reads onto the genome to identify peak regions presumably corresponding to TFBS ((Figure 7A). The sites in TFBS are expected to be represented more frequently than those in the flanking sequences (Figure 7B). This leads to an approximate location of TFBS. If multiple non-homologous genes share the seven sites in red (Figure 7A), but not the flanking sequences, then we may infer that the seven sites constitute a TFBS for the particular TF of interest.

ChIP-seq also has the shortcoming of spurious cross-linking and large sample size which is needed for us to see the peak in Figure 7. We would not need a large sample of cells (i.e., genomes) if we could trim off all flanking sequences from the TFBS-containing DNA fragments. This is one of the reasons that newer methods such as ChIP-exo [17,18], ChEC-seq [19] and Cut&Run [20] require fewer cells. These newer methods also do not need sonication. Another promising method is DamID-seq [21] which only requires the construction of a fusion protein and methylation-specific genome sequencing, and theoretically could work with single cells.

The second method of extracting TFBS is by Gibbs sampler which has been used in the identification of functional motifs in proteins [89,90,91] and regulatory sequences in DNA [92,93,94,95,96,97,98,99,100]. The algorithm has been numerically illustrated in detail [62,101] and implemented in DAMBE [66,67]. It is particularly useful for extracting TFBS when we have few reads mapped to each gene or when there are few cells to start with. Figure 8A shows six reads each mapped to a separate gene. We know that a motif may be hidden in these reads, but do not know what the motif looks like or where it is located in the reads. Gibbs sampler is exactly suitable for identifying this motif and pinpointing its location. One may compile the sequences into a file (Figure 8B) and feed it to Gibbs sampler. The typical output includes a PWM (Figure 8C) that one can use to scan the genome for new locations of the TFBS, and a motif that is over-represented in the input sequences (Figure 8D). With new technologies such as Cut&Run [20] that is touted to work with few cells, I expect Gibbs sampler to become an essential tool embedded in all bioinformatics software packages used for protein-DNA interactions.

Some researchers (e.g., [53]) thought that ChIP-seq experiments do not inherently measure the relative motif preference of a TF to individual sequences, but they do just that, as is illustrated in Figure 8. Given the PWM in Figure 8C, the preference for TACGTGT is measured by the PWM score (PWMS) and is equal to 13.06491 (the summation of the bolded numbers). Another 7mer, TACGCGT, resulting from a transition at the 5th site, would have a PWMS of 11.75622. Thus, the relative motif preference of a TF can be readily quantified.

## 4. Conclusions

Regulon research is a rapidly progressing field with many new experimental protocols, computational algorithms, and databases having been developed in recent years. I present conceptual rationales of these protocols and algorithms and highlight their advantages and disadvantages. Existing literature on regulon research, especially in tree species, were critically reviewed to highlight how better results could have been obtained.

## Figures and Tables

**Figure 1 genes-11-00995-f001:**
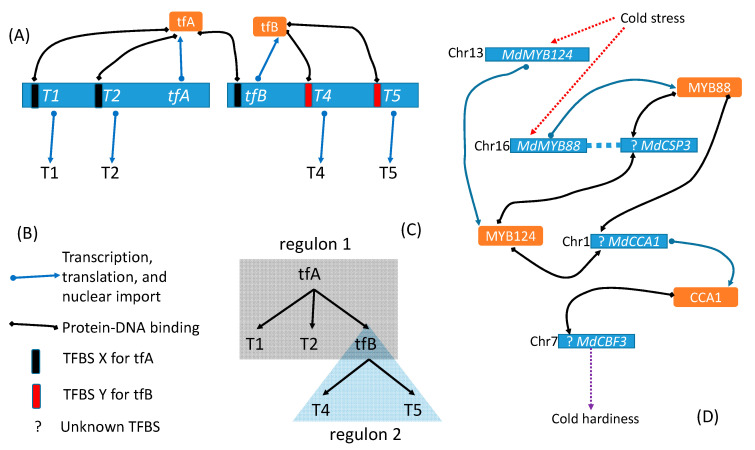
Illustration of a regulon network of two interacting regulons. (**A**) tfA-regulon with target genes *T1*, *T2*, and *tfB*, and tfB-regulon with target genes *T4* and *T5*. (**B**) Legends of graphic elements. TFBS—Transcription factor binding site. (**C**) A simplified representation of the two regulons. (**D**) A partial regulon network [3] for coping with cold stress in apple (*Malus demestica*), in which TFBSs remain poorly characterized.

**Figure 2 genes-11-00995-f002:**
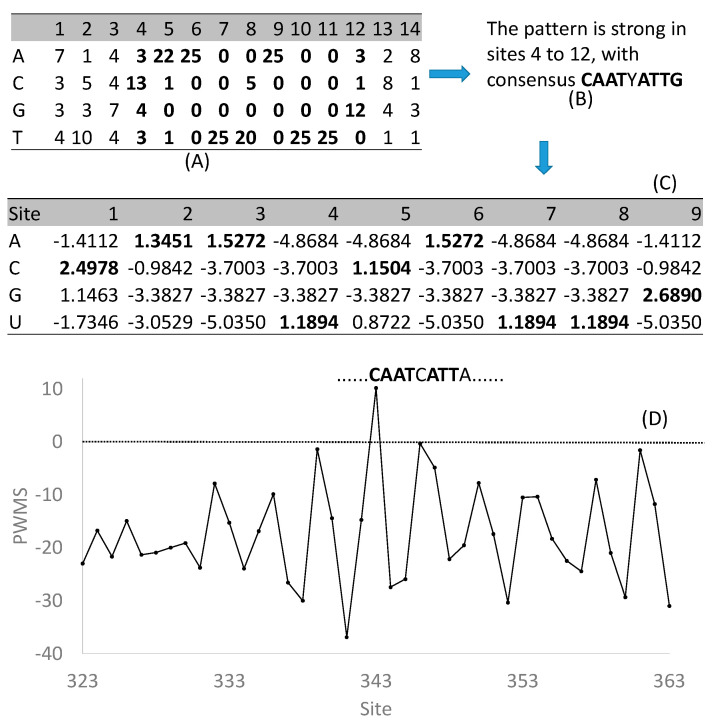
Construction and use of position weight matrix (PWM) to discover new transcription factor binding sites (TFBS). (**A**) Site-specific compilation of experimentally validated TFBS for AtHB1, retrieved from PlantPAN [57]. (**B**) The consensus sequence from nine strongly informative sites in (**A**). (**C**) PWM obtained from data in (**A**). (**D**) PWM score (PWMS) for 9mers along the 2000 nt upstream of micro-RNA gene *miR164a* which starts at site 2001. The expected PWMS for random sequences is 0. The 9mer with the highest PWMS is CAATCATTA (at Site 343) which was verified to be an effective TFBS for AtHB1. PWM computation and sequence scanning were done with software DAMBE [66,67].

**Figure 3 genes-11-00995-f003:**
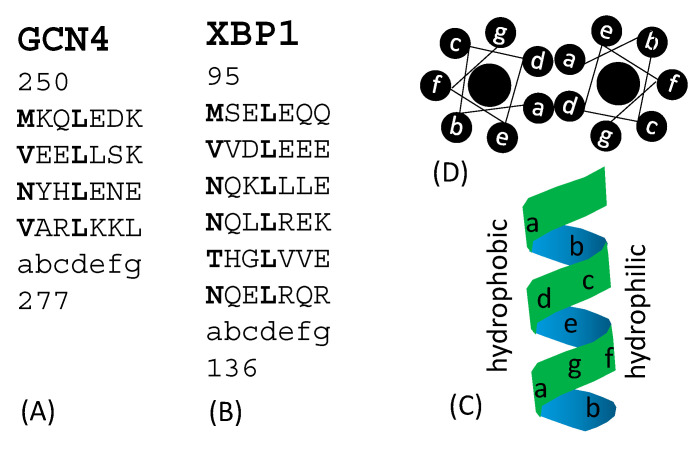
Examples of leucine zipper transcription factors with heptad repeats, with the seven amino acids in each heptad labelled as abcdefg. Hydrophobic residues at positions a and d are shown in bold (**A**) GCN4 (sites 250 to 277) in yeast *Saccharomyces cerevisiae.* (**B**) XBP1 (sites 95 to 136) in human. (**C**) Relative position of the seven amino acids in an α helix. (**D**) Top view of a leucine zipper homodimer contacting at their hydrophobic side.

**Figure 4 genes-11-00995-f004:**
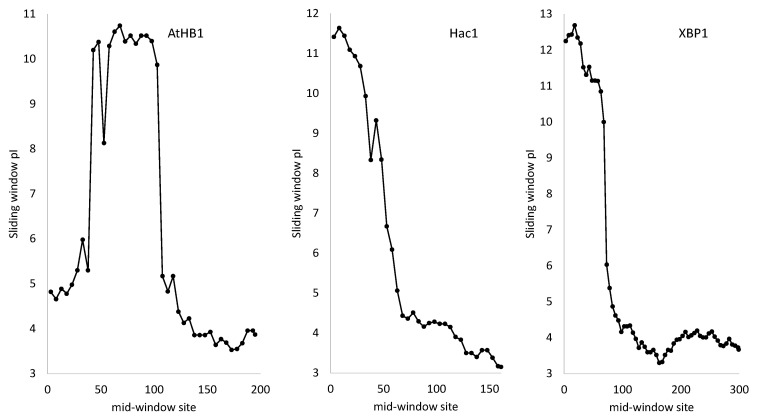
Protein isoelectric point (*pI*) profile showing different transcription factors having a positively charged DNA-binding domain, with AtHB1 from *Arabidopsis thaliana,* Hac1 from *Saccharomyces cerevisiae* and XBP1 from human (a functional homologue of Hac1). Window-specific *pI* was computed with software DAMBE [66,67] with window size of 80 and step size of 5.

**Figure 5 genes-11-00995-f005:**
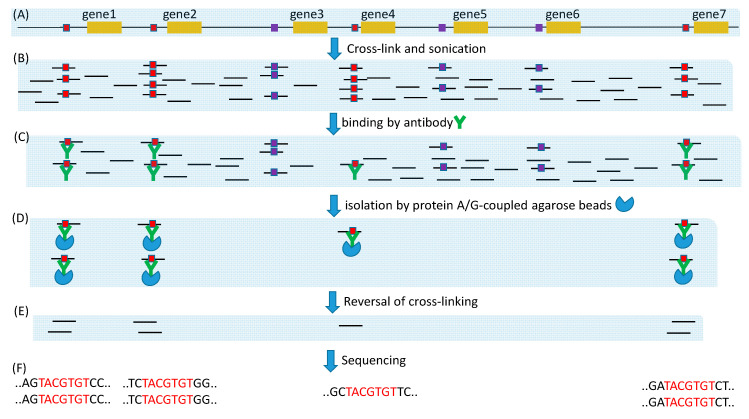
Illustration of ChIP-Seq method for highlighting its problems and improvements. (**A**) DNA genome with gene1, gene2, gene 4, and gene7 sharing the same TFBS bound to the same TF of interest (red-colored squares). (**B**) Cross-linking fixes TF onto TFBS, and sonication is optimized to minimize the sequences flanking TFBS. (**C**) A specific anti-TF antibody is used to bind to the TF of interest. (**D**) A general-purpose protein A/G coupled to agarose beads is used to pull down the TFBS- TF- antibody complex. (**E**) Reversal of cross-linking frees TFBS-containing fragments for sequencing. (**F**) High-throughput sequence generates millions or even billions of sequence reads that may contain a TFBS.

**Figure 6 genes-11-00995-f006:**
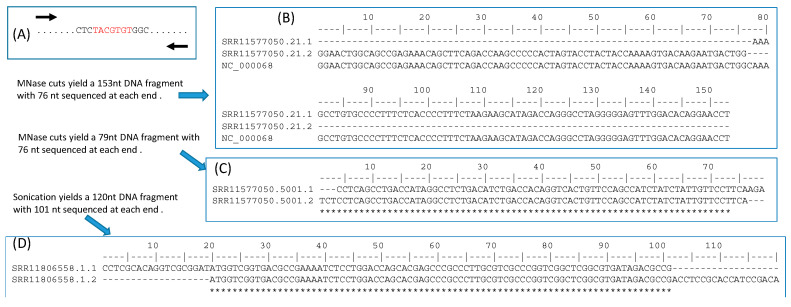
Reducing background sequences flanking TFBS. (**A**) Paired-end reading of a TFBS- containing sequence fragment. (**B**) Paired-end reads that do not overlap, from an SRA file (ACCN: SRR11577050 in BioProject PRJNA544746) generated by Cut&Run. (**C**) Paired-end reads overlap, from the same file as in (**B**). (**D**) Paired-end reads from an SRA file (ACCN: SRR11806558 in BioProject PRJNA633509) generated by ChIP-seq. The forward or reverse reads is reverse-complemented to generate the sequence alignment.

**Figure 7 genes-11-00995-f007:**
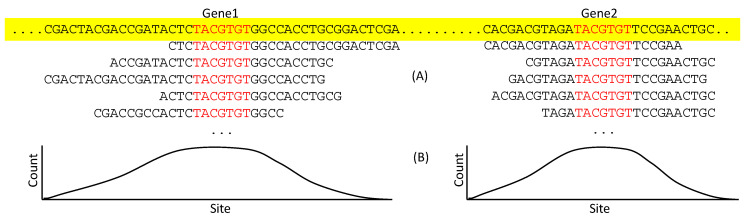
Identifying TFBS by mapping ChIP-Seq reads to genome. (**A**) Two genes on the genome (colored yellow) are not homologous but share a TFBS (colored red), each with many reads mapped to them. (**B**) The count of each site represented in reads, with each site in TFBS expected to have higher counts than flanking sequences.

**Figure 8 genes-11-00995-f008:**
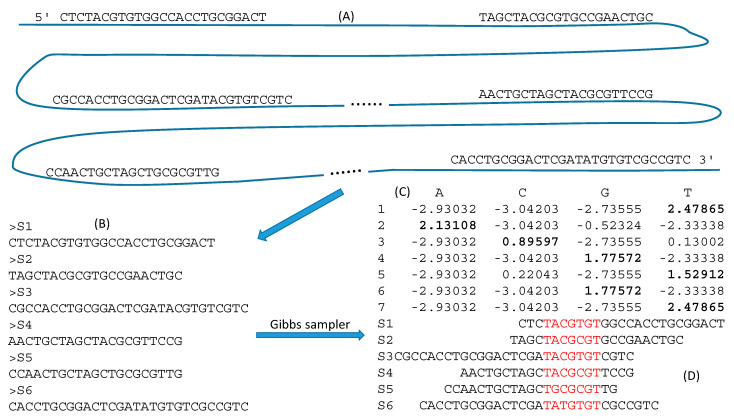
Using Gibbs sampler to extract TFBS. (**A**) Six reads mapped each mapped to a different gene on the genome (represented by the blue line. (**B**) FASTA file for the six reads as input to Gibbs sampler. (**C**) An optimal position weight matrix (PWM) that one can use to scan the genome for new locations of the TFBS. (**D**) The aligned motif that generates the most informative PWM in (**C**). (**C**) and (**D**) are generated from DAMBE [61] based on the FASTA input file in (**B**).

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
