# Peer review of "Beyond Trees: Regulons and Regulatory Motif Characterization"

_genes, 2020, doi:10.3390/genes11090995_

Round 1

Reviewer 1 Report

The manuscript describes several concepts on the matter of transcription factor-DNA interactions and the properties of networks they can interact with. Specifically they define transcription factors and networks and the computational approaches for their discovery. This includes one of the only and in my experience the best explanation of binding site motifs by PWM (position weight matrix). There is also a section on the analyses of protein acidity to detect DNA binding proteins. The manuscript is superbly written, both for general clarity and the ability to convey computational approaches to non-experts.

The title implies the manuscript is in some way about trees: “Beyond trees: regulons and regulatory motif characterization.” It is not other than to say that not much has happened in trees. The topic of trees should be removed from the manuscript unless the author would like to include that into the subject matter. If there is some knowledge of tree regulons or sequence motifs then that should be mentioned along with a road map to building the necessary resources and propose approaches that will be necessary to arcane tree genomics. If there is truly nothing to review in trees then that topic should be removed from the review.

They make broad claims about the field not advancing, but only cite a narrow set of literature, mostly their own. It is difficult to classify it as a review in that it does not describe any one area in any depth. The document appears to be written around the author's body of literature to the point where topics are included in order to include a citation to their publications. For example, there is very little reason to mention microarrays. Readers need not understand this largely outdated technology to describe the future of tree computational genomics or to review the field.

While I do appreciate the explanation of each computational approach, more approaches should be discussed. What are the pros and cons of regulon analysis versus other representations of regulatory networks, weighted coexpression analysis, for example? They provide a limited exploration of the challenges after finding motifs, i.e., does the presence of the motif actually confer useful information, why are there so many instances of the motif genome wide, do motifs in coding regions, introns, UTR’s have a role? How does chromatin accessibility influence motif discovery? Are there any databases of open chromatin for trees? 

I would like to bring three references to the attention of the author. There are many relevant citations that I could mention, but these in areas that should be included or expanded in this review.

Weirauch, Matthew T. et al. Determination and Inference of Eukaryotic Transcription Factor Sequence Specificity Cell, Volume 158, Issue 6, 1431 - 1443.

O’Malley, R. C., Huang, S.-S. C., Song, L., Lewsey, M. G., Bartlett, A., Nery, J. R., et al. (2016). Cistrome and Epicistrome Features Shape the Regulatory DNA Landscape. Cell, 165(5), 1280–1292. 

Kulkarni, S. R., Jones, D. M., & Vandepoele, K. (2019). Enhanced Maps of Transcription Factor Binding Sites Improve Regulatory Networks Learned from Accessible Chromatin Data. Plant Physiology, 181(2), 412–425. 

Other comments

L26:27. Reference to arginine biosynthesis but no mention of what organism, adding E. coli to the sentence would help with readability - E. coli mentioned later in the paragraph. 

L46. Citation of own paper about analysis of microarray data from 2001 seems unnecessary when trying to say that motif characterization hasn’t kept up with the times

3.1 Regulon discovery by checking regulon dictionary

The JASPAR TFDB should be included in the list of databases.

Addition of useful tools for whole genome PWM scans is suggested - specifically FIMO, HOMER, and PWMScan https://ccg.epfl.ch/pwmscan/. 

L203:206. That PIF1/PIL5 has been shown to regulate but doesn’t show direct binding would seem to indicate the need for a regulatory network approach as opposed to individual regulons. The information that PIF1/PIL5 can regulate AtHB1 (directly or indirectly) is lost if you are limited to the regulon approach. 

3.2 De novo regulon discovery

Figure 3 doesn’t seem to serve much purpose in a review about tree regulons - except to mention the author’s own software. In the author’s own words “The features shared by transcription factors unfortunately are not sufficient for transcription factor or TFBS identification through bioinformatics means.” - given this, the paragraph is not well justified. 

When discussing the techniques involved in identifying transcription factor binding it would seem worth the effort to include modern updates to ChIP-SEQ such as ChIP-exo and ChIP-nexus, or μ-chip, and their improvements to the chip-seq approach. DAP-seq should be mentioned as a modern high throughput in vitro technique for identifying putative binding sites, and motifs. In truth the DAP-seq method seems ideal for discovering possible regulons. 

No mention of antibodies in the chip-seq approach section. A brief description of chip-seq results being dependent on antibody quality and the possible use of tagged proteins is suggested. 

L266:267. A discussion of the CUT&RUN technique and/or DamID-seq is suggested to provide context for transient tf binding.   

When discussing methods of motif identification from chip-seq samples there needs to be a broader exploration of the tools available. The MEME-suite, MACS2, and HOMER softwares are standards and need to be mentioned at a minimum. A reference to the ENCODE project and ChIP-SEQ guidelines is also suggested. 

When discussing motif finding algorithms the original Gibbs sampler paper (Lawrence et al. Science 1993) should be referenced. A brief mention of alternative motif finding algorithms - ie. enumerative/deterministic (see PMCID: PMC6490410 for a recent review) is highly recommended. 

Figure 4 implies that after aligning to a genome and finding peaks that the peak = the motif. This is misleading, after peak finding another step should be shown where peaks are put through a motif finding algorithm to search for motif enrichment. 

May be useful to mention tree specific genomic resources - ie. PopGenIE, EucGenIE, TreeGenesDB, Phytozome and how they can be used for regulon discovery.

Author Response

Dear Editor,

Thank you for securing excellent reviews that revealed multiple weaknesses of the paper. My effort to address these weaknesses has resulted in significant improvement of the manuscript. I detail my point-by-point responses below in Bold italic.

Comments and Suggestions for Authors (Review #1)

The manuscript describes several concepts on the matter of transcription factor-DNA interactions and the properties of networks they can interact with. Specifically they define transcription factors and networks and the computational approaches for their discovery. This includes one of the only and in my experience the best explanation of binding site motifs by PWM (position weight matrix). There is also a section on the analyses of protein acidity to detect DNA binding proteins. The manuscript is superbly written, both for general clarity and the ability to convey computational approaches to non-experts.

The title implies the manuscript is in some way about trees: “Beyond trees: regulons and regulatory motif characterization.” It is not other than to say that not much has happened in trees. The topic of trees should be removed from the manuscript unless the author would like to include that into the subject matter. If there is some knowledge of tree regulons or sequence motifs then that should be mentioned along with a road map to building the necessary resources and propose approaches that will be necessary to arcane tree genomics. If there is truly nothing to review in trees then that topic should be removed from the review.

Yes, this paper is for a special issue on "Tree genetics". I have added some critical evaluation of some research on tree regulon research.

They make broad claims about the field not advancing, but only cite a narrow set of literature, mostly their own. It is difficult to classify it as a review in that it does not describe any one area in any depth. The document appears to be written around the author's body of literature to the point where topics are included in order to include a citation to their publications. For example, there is very little reason to mention microarrays. Readers need not understand this largely outdated technology to describe the future of tree computational genomics or to review the field.

Many more relevant papers were included in the review and the number of figures has increased from 4 to 8. I have reviewed de novo discovery of transcription factors (TF) as well as of transcription factor binding sites (TFBS). The figure including an illustration of microarray methods in characterizing transcription factor binding sites is removed.

In the section on de novo discovery of TF, I have included an illustration of easily recognizable structural features in certain classes of TF such as leucine zipper with characteristic heptad repeats (after learning that many of our graduate students in molecular biology do not know what heptad repeats are).

While I do appreciate the explanation of each computational approach, more approaches should be discussed. What are the pros and cons of regulon analysis versus other representations of regulatory networks, weighted coexpression analysis, for example? They provide a limited exploration of the challenges after finding motifs, i.e., does the presence of the motif actually confer useful information, why are there so many instances of the motif genome wide, do motifs in coding regions, introns, UTR’s have a role? How does chromatin accessibility influence motif discovery? Are there any databases of open chromatin for trees?

These are all excellent questions that one need to address in a study of regulons. I have included a discussion of a BioProject (PRJNA633509) in the NCBI SRA database to address some of these problems. The project is to characterize transcription factor narrow sheath1 (NS1), its target genes, and its binding sites in Zea mays. The ChIP-Seq data set would allow one to identify putative TFBSs in a large number of putative genes. If genes A, B, C harbors the TFBS, then one would predict that genes A, B, C would exhibit different expression in cells with NS1 present from cells with NS1 absent. The BioProject included transcriptomic data from the wild type with a function NS1 and from an ns1 mutant. Thus, if genes A, B, and C not only harbors a TFBS, but also expressed differently in cells with a functional NS1 from cells with NS1 knocked out, then one is fairly confidently conclude that genes A, B, and C are target genes of NS1. If only genes A and B behave in this way but gene C does not (i.e., it harbors a TFBS but its expression does not change with presence or absence of NS1), then there must be other factors contributing to the expression of gene C, etc.

An alternative confirmation approach is to compare sequence conservation of orthologous genes among related species. If, among orthologues of gene C, the putative TFBS is not conserved, then it perhaps no longer functions as a TFBS. In constrast, if the TFBS is far more conserved than its flanking sequences, then it would suggest some alternative hypotheses.

Ideally, one should have not only just ChIP-seq and RNA-seq data, but also a profile of open chromatin. A TF failing to regulate one of its target genes may be caused by the target gene not in open chromatin region, so the TFBS of the target gene is not available for binding by the TF.

In short, the reviewer is right in pointing out that there are many related questions in a regulon study. I have tried to address as many questions as I can without deviating too far from promoting regulon characterization in tree genetics.

I would like to bring three references to the attention of the author. There are many relevant citations that I could mention, but these in areas that should be included or expanded in this review.

Weirauch, Matthew T. et al. Determination and Inference of Eukaryotic Transcription Factor Sequence Specificity Cell, Volume 158, Issue 6, 1431 - 1443.

O’Malley, R. C., Huang, S.-S. C., Song, L., Lewsey, M. G., Bartlett, A., Nery, J. R., et al. (2016). Cistrome and Epicistrome Features Shape the Regulatory DNA Landscape. Cell, 165(5), 1280–1292.

Kulkarni, S. R., Jones, D. M., & Vandepoele, K. (2019). Enhanced Maps of Transcription Factor Binding Sites Improve Regulatory Networks Learned from Accessible Chromatin Data. Plant Physiology, 181(2), 412–425.

These references are indeed highly relevant as they all represent efforts to characterize TFBS. They are included. I do have some reservation on Weirauch et al. From the observation that "closely related DBDs almost always have very similar DNA sequence preferences", they jump to the conclusion that "in general, sequence preferences can be accurately inferred by overall DBD AA identity". I do not disagree with the observation that "closely related DBDs almost always have very similar DNA sequence preferences" and can list examples more impressive than they did. For example, TFs such as Hac1p in yeast, Xbp1 in mammals and BZIP60 in plants all have a leucine zipper DNA-binding structure, and their TFBS all feature the core ACGT. However, I disagree that such observations, remarkable as they are, imply much power in making specific predictions. For example, if I state that "Because transcription factor X has a DBD nearly identical to that of transcription factor Y, I predict that transcription factor X has nearly identical DNA sequence preference as that of transcription factor Y", few experimental biologists would take my prediction seriously and act upon it. In other words, few people would agree with their assertion that "in general, sequence preferences can be accurately inferred by overall DBD AA identity". One can list counter examples in which sequence preferences cannot be accurately inferred by overall DBD AA identity.

The database they built (http://cisbp.ccbr.utoronto.ca/) includes TFBS from some large-scale studies, but not TFBS detected and verified through conventional painstaking experiments. For example, it includes the second TFBS of yeast Hac1p, but ignores the first and well-characterized TFBS (GACAGCGTGTC). The same is true for database JASPAR. Such incomplete database could be misleading. If a new researcher identified GACAGCGTGTC as a TFBS for Hac1p, he may search the databases for Hac1p TFBS. Upon finding no such an entry, he may be misled to think that he has identified a TFBS new to science. This being said, I do find the databases highly valuable.

I have highlighted the value of O'Malley et al, especially in the context of lowly expressed or transiently expressed TF whose TFBS would be hard to characterize without having the TF highly expressed in vitro.

I was interested in the paper by Kulkarni et al. for one particular reason (especially the part of the title "accessible chromatin data"). When one scans the DNA region upstream of coding sequences (CDSs) for TFBS of a TF, one often has gene A harboring a putative TFBS with a higher position weight matrix score (PWMS) than gene B. However, gene B expression responds to the TF expression more consistently than gene A. One possible reason is that gene B's TFBS is in accessible chromatin when the TF is expressed, but gene A's TFBS is not in accessible chromatin when the TF is expressed. Thus, although gene A's TFBS has a high PWMS and is supposedly a strong signal for TF, it is not regulated by the TF because the TF cannot access it. However, the paper does not address such specific problems that we are interested in.

Other comments

L26:27. Reference to arginine biosynthesis but no mention of what organism, adding E. coli to the sentence would help with readability - E. coli mentioned later in the paragraph.

Revised.

L46. Citation of own paper about analysis of microarray data from 2001 seems unnecessary when trying to say that motif characterization hasn’t kept up with the times

Removed together with the part on microarray.

3.1 Regulon discovery by checking regulon dictionary

The JASPAR TFDB should be included in the list of databases.

I included JASPAR but not TFDB (which is specifically for mouse and is consequently not relevant for a paper in a special issue on "Tree genetics")

Addition of useful tools for whole genome PWM scans is suggested - specifically FIMO, HOMER, and PWMScan https://ccg.epfl.ch/pwmscan/.

Cited.

L203:206. That PIF1/PIL5 has been shown to regulate but doesn’t show direct binding would seem to indicate the need for a regulatory network approach as opposed to individual regulons. The information that PIF1/PIL5 can regulate AtHB1 (directly or indirectly) is lost if you are limited to the regulon approach.

A good point. highlighted.

3.2 De novo regulon discovery

Figure 3 doesn’t seem to serve much purpose in a review about tree regulons - except to mention the author’s own software. In the author’s own words “The features shared by transcription factors unfortunately are not sufficient for transcription factor or TFBS identification through bioinformatics means.” - given this, the paragraph is not well justified.

The figure is removed.

When discussing the techniques involved in identifying transcription factor binding it would seem worth the effort to include modern updates to ChIP-SEQ such as ChIP-exo and ChIP-nexus, or μ-chip, and their improvements to the chip-seq approach. DAP-seq should be mentioned as a modern high throughput in vitro technique for identifying putative binding sites, and motifs. In truth the DAP-seq method seems ideal for discovering possible regulons.

I have discussed DAP-seq in the context of lowly expressed or transiently expressed TF in which few TF-TFBS complexes exist so in vitro high expression of the TF is essential. I have discussed the advantages of ChIP-exo, ChIP-nexus (which is an extension of ChIP-exo), ChEC-seq, Cut&Run after presenting the workflow of ChIP-Seq. I have also illustrated the application of Gibbs sampler to protocols with limited number of genomes.

I did not include mChIP. There are two technologies using this name, one is large-scale as an extension of microarray technology, and the other is small-scale associated with PCR detection. They are probably already obsolete.

No mention of antibodies in the chip-seq approach section. A brief description of chip-seq results being dependent on antibody quality and the possible use of tagged proteins is suggested.

Included in a figure.

L266:267. A discussion of the CUT&RUN technique and/or DamID-seq is suggested to provide context for transient tf binding.  

Included.

When discussing methods of motif identification from chip-seq samples there needs to be a broader exploration of the tools available. The MEME-suite, MACS2, and HOMER softwares are standards and need to be mentioned at a minimum. A reference to the ENCODE project and ChIP-SEQ guidelines is also suggested.

Gibbs sampler is an algorithm with many manifestations. It can work with a large number of vectors or a large number of strings. MEME-suite is a set of programs of which the one used for motif discovery implments Gibbs sampler. The method represented by MACS was mentioned before but MACS was not cited. I have added Figure 7 to illustrate its conceptual rationale. HOMER is cited (its core is really just a scanner based on PWM).

When discussing motif finding algorithms the original Gibbs sampler paper (Lawrence et al. Science 1993) should be referenced. A brief mention of alternative motif finding algorithms - ie. enumerative/deterministic (see PMCID: PMC6490410 for a recent review) is highly recommended.

Lawrence et al. Science 1993 was in fact cited in the very first version of the manuscript.

I found the review (MCID: PMC6490410) rather poorly written. There should be just two categories of methods, one without using information of site-dependence and one does (high-order Markov models, not Markov chain models. The authors of the review did not make a difference).

Figure 4 implies that after aligning to a genome and finding peaks that the peak = the motif. This is misleading, after peak finding another step should be shown where peaks are put through a motif finding algorithm to search for motif enrichment.

Figure 4 is replaced with Figures 5-8.

May be useful to mention tree specific genomic resources - ie. PopGenIE, EucGenIE, TreeGenesDB, Phytozome and how they can be used for regulon discovery.

The revised manuscript has become quite long, so I chose not to include these because they are not quite related to the theme of promoting regulon research in tree genetics.

Thank you again for handling the manuscript.

Sincerely,

Xuhua Xia

Professor of Biology

Reviewer 2 Report

     In this article the authors described experimental and bioinformatics approaches for characterizing the transcription factors binding sites and discovering of new plant regulons. Using various examples, they explained the term 'regulon' and the rules of operation of the regulon networks.

     The manuscript is well presented and easy to follow. The organization of the different sections is adequate and the headings always reflect their content. However, I feel unsatisfied with examples of how regulons work in trees. This greatly reduces the value of this article.

Furthermore some aspects of the manuscript should be improved (see comments below).

1. The point 2A should be changed to point 2.

2. The experimental data included in this review should not be based on the other review paper but on the original papers (page 2, line 81). Please change!

3. In the seventh paragraph of chapter 2 (page 3, lines 102 ) the authors indicate that some regulons in tree species are well characterized. However, they did not give any examples of such regulons, as well as they did not characterize their mode of operation.

4. 47 citations from 74 references were published more than 10 years ago. There are a lot of current publications that are relevant in such a hot area. Please check current ones, if available!5. The section “Conclusions” is one of the weakest parts of this article and adds nothing to the entire article.        

     In general the review is a short overview of the available literature focusing on the effects of regulons (based on selected examples). The authors also made an attempt to characterize the methods of de novo regulon detection. However, this article has no key teaching point or hypothesis that the reader can take from it. Moreover, as I mentioned earlier, I am not satisfied with information on the use of transcription factor databases to aid the characterization of tree regulons.

Author Response

Dear Editor,

Thank you for securing excellent reviews that revealed multiple weaknesses of the paper. My effort to address these weaknesses has resulted in significant improvement of the manuscript. I detail my point-by-point responses below in Bold italic.

Reviewer #2

In this article the authors described experimental and bioinformatics approaches for characterizing the transcription factors binding sites and discovering of new plant regulons. Using various examples, they explained the term 'regulon' and the rules of operation of the regulon networks.

     The manuscript is well presented and easy to follow. The organization of the different sections is adequate and the headings always reflect their content. However, I feel unsatisfied with examples of how regulons work in trees. This greatly reduces the value of this article.

The point is well taken. I have added a number of new examples for illustrating the underlying conceptual framework.

Furthermore some aspects of the manuscript should be improved (see comments below).

  1. The point 2A should be changed to point 2.

I can't find point 2A. Is it Figure 2A?

  1. The experimental data included in this review should not be based on the other review paper but on the original papers (page 2, line 81). Please change!

Changed.

  1. In the seventh paragraph of chapter 2 (page 3, lines 102 ) the authors indicate that some regulons in tree species are well characterized. However, they did not give any examples of such regulons, as well as they did not characterize their mode of operation.

I have added examples and illustrated the apple MdMYB88 and MdMYB124 regulon in Figure 1 (The two are close paralogues with nearly identical AA sequences and regulate the same set of genes)

  1. 47 citations from 74 references were published more than 10 years ago. There are a lot of current publications that are relevant in such a hot area. Please check current ones, if available!5. The section “Conclusions” is one of the weakest parts of this article and adds nothing to the entire article.

I have integrated many new references (from 74 to 101), added five new figures, and revised the conclusion. Some old references (e.g., Xia and Xie 2001. Bioinformatics) were removed. However, some other reviewers insist on citing some old referneces.

     In general the review is a short overview of the available literature focusing on the effects of regulons (based on selected examples). The authors also made an attempt to characterize the methods of de novo regulon detection. However, this article has no key teaching point or hypothesis that the reader can take from it. Moreover, as I mentioned earlier, I am not satisfied with information on the use of transcription factor databases to aid the characterization of tree regulons.

It is now much longer with 8 figures and 101 references.

Thank you again for handling the manuscript.

Sincerely,

Xuhua Xia

Professor of Biology

Reviewer 3 Report

The manuscript entitles “Beyond tress: regulons and regulatory motif characterization” comprehensively describes a regulon and methods to find regulons. This review is well-written, particularly, about ways that find a transcriptional binding motif.  I believe that this review fascinates a broad range of researchers interested in gene expression regulation. However, I do think there are some things that should be addressed to improve this paper.

Major points

1)   Despite the title contains the word “trees”, there are few descriptions as to tree regulons. I would like to know recent findings of a regulon of CO-TF in trees, if at all possible.

2) Please describe a difference between a gene regulatory network (GRN) and a regulon network, if these are different. Even if these are similar meanings, please explain it, since a GRN is more commonly used than a regulon network.

3)   The author should mention DAP-Seq in section 3.2. Since ChIP followed by microarray method (ChIP-chip) seems to be a past technique, I would recommend mentioning DAP-Seq instead of ChIP-chip.

Minor point

Line 93-101. I may disagree with a logic that a robust regulon network represented as a tree should have low height, because, for instance, an incoherent feedforward loop reduces noise even though it has an additional component comparing with a straight regulatory network. I also cannot entirely agree that reported gigantic gene regulation networks are usually artifacts.

Author Response

Dear Editor,

Thank you for securing excellent reviews that revealed multiple weaknesses of the paper. My effort to address these weaknesses has resulted in significant improvement of the manuscript. I detail my point-by-point responses below in Bold italic.

Reviewer #3

The manuscript entitles “Beyond tress: regulons and regulatory motif characterization” comprehensively describes a regulon and methods to find regulons. This review is well-written, particularly, about ways that find a transcriptional binding motif.  I believe that this review fascinates a broad range of researchers interested in gene expression regulation. However, I do think there are some things that should be addressed to improve this paper.

Major points

1)   Despite the title contains the word “trees”, there are few descriptions as to tree regulons. I would like to know recent findings of a regulon of CO-TF in trees, if at all possible.

I have added more examples of tree regulons, and illustrated one such regulon network in Figure 1.

2) Please describe a difference between a gene regulatory network (GRN) and a regulon network, if these are different. Even if these are similar meanings, please explain it, since a GRN is more commonly used than a regulon network.

Explained. A regulon network (RN) differs from the conventional gene regulation network (GRN) mainly in the basic unit of the network. A basic unit in RN is a regulon with its four features (TF, TFBS, target genes and biological function), but that in a GRN is not explicitly defined, although two nodes in GRN connected by a directed or undirected line are assumed to be in regulator-regulatee relationship. An RN highlights which feature of a regulon is missing.

3)   The author should mention DAP-Seq in section 3.2. Since ChIP followed by microarray method (ChIP-chip) seems to be a past technique, I would recommend mentioning DAP-Seq instead of ChIP-chip.

Yes, I have highlighted the value of DAP-seq especially in the context of lowly expressed or transiently expressed TF in which TF-TFBS complexes are rare. High TF expression in vitro dramatically increases the number of TFBSs that can be characterized.

Minor point

Line 93-101. I may disagree with a logic that a robust regulon network represented as a tree should have low height, because, for instance, an incoherent feedforward loop reduces noise even though it has an additional component comparing with a straight regulatory network. I also cannot entirely agree that reported gigantic gene regulation networks are usually artifacts.

Comments well taken. My point is that a chain with 100 links is not as robust as a chain with 10 links. Of course information propagation in biology is rarely through a single chain but more often through redundant pathways to increase robustness. I have rephrased.

Thank you again for handling the manuscript.

Sincerely,

Xuhua Xia

Professor of Biology